

# Study on the applicability of microtremor HVSR method to support seismic microzonation in the town of Idrija (W Slovenia)

Andrej Gosar[1,2]

[1]University of Ljubljana, Faculty of Natural Sciences and Engineering, Ljubljana, SI-1000, Slovenia
[2] Slovenian Environment Agency, Seismology and Geology Office, Ljubljana, SI-1000, Slovenia

*Correspondence to*: Andrej Gosar (andrej.gosar@gov.si)

**Abstract.** The Idrija town is located in area with increased seismic hazard in W Slovenia and is partly built on alluvial sediments or artificial mining and smelting deposits which can amplify seismic ground motion. There is a need to prepare a comprehensive seismic microzonation in the near future to support seismic hazard and risk assessment. To study the applicability of microtremor Horizontal-to-Vertical Spectral Ratio (HVSR) method for this purpose, 70 free-field microtremor measurements were performed in 0.8 km$^2$ large town area with 50-200 m spacing between points. HVSR analysis has shown that it is possible to derive sediments resonance frequency at 48 point, whereas at remaining one third of measurements nearly flat HVSR curves were obtained indicating small or no impedance contrast with the seismological bedrock. Iso-frequency (range 2.5-19.5 Hz) and HVSR peak amplitude (range 3-6, with few larger values) maps were prepared by using natural neighbour interpolation algorithm and compared with the geological map and map of artificial deposits. Surprisingly no clear correlation was found between distribution of resonance frequencies or peak amplitudes and the known extent of supposed "soft" sediments or deposits. This can be explained by relatively well compacted and rather stiff deposits and complex geometry of sedimentary bodies. However, at several individual locations it was possible to correlate the shape and amplitude of the HVSR curve with the known geological structure and prominent site effects were established in different places. On the other hand, in given conditions (very limited free space and high level of noise) it would be difficult to perform active seismic refraction or MASW measurements to investigate the S-waves velocity profiles and thickness of sediments in details, which would be representative enough for microzonation purposes. The importance of microtremor method is therefore even greater, because it enables direct estimation of the resonance frequency without knowing the internal structure and physical properties of the shallow subsurface. The results of this study can be used directly in analyses of possible occurrence of soil-structure resonance of individual buildings, including important cultural heritage mining and other structures protected by UNESCO. Second application of the derived free-field iso-frequency map is to support soil classification according to the recent trends in building codes.



## 1 Introduction

The town of Idrija (8.000 inhabitants) is an important industrial and cultural centre in the western Slovenia (Fig. 1a), for which the increased seismic hazard is characteristical. It is located in relatively narrow valleys at the confluence of Nikova and Idrijca rivers (Fig. 1b). As the oldest mining town in Slovenia, it is famous for more than 500 years of mercury mining
tradition. Part of the town is built on artificial deposits which resulted from mining and smelting activities or on alluvial sediments. However, the extent, thickness and stiffness of artificial deposits are partly unknown, because they were accumulated through centuries of mining activities (Gosar and Čar, 2006). For this kind of relatively soft deposits or sediments, it is usual that they amplify seismic ground motion during an earthquake, that is known as seismological site effects (Reiter, 1990). However, no seismic microzonation of the Idrija urban area was performed so far, which would
quantitatively assess effects of the local geological structures on seismic ground motion. Therefore, there is a great necessity to prepare a comprehensive seismic microzonation in a near future to support seismological hazard and risk assessment, as well as to provide input data for earthquake resistant design of buildings and structures or retrofitting of existing buildings. This is especially important because Idrija is an important centre which undergoes fast development. In addition it is an UNESCO world heritage site in which a very important industrial and cultural heritage buildings and structures related to
500 years of mercury mining should be protected from all kinds of natural hazards including the seismic hazard.

Methodologies used in the seismic microzonation has been significantly improved in the last two decades through better understanding of seismological site effects, development of new research methods for characterization of the shallow subsurface and through preparation of appropriate standards and guidelines (Ansal, 2004). Seismic microzonation is usually
performed at different levels, from a preparatory level (level 1) which is based on existing data, trough intermediate level (level 2) in which simplified methods and empirical laws are used to assess ground motion amplification, to the highest level (level 3) which involves advanced methods for quantitative characterisation of the sediments and application of numerical methods to derive amplification factors (Bramerini et al., 2015). Level 3 seismic microzonation provides input data for the state-of-the-art earthquake resistant design or retrofitting. Among other geophysical and geotechnical methods used for the
level 3 microzonation, seismological microtremor HVSR (Horizontal-to-Vertical Spectral Ratio) method has achieved high recognition in the last decade, because it provides very important data on the main resonance frequency of soft sediments overlaying stiff geological bedrock (Bard, 1999). Results of the free-field microtremor measurements are thus used in different soil classification standards and especially valuable when compared with additional microtremor measurements inside buildings to assess the danger of soil-structure resonance which can significantly enhance the damage during an
earthquake (Gallipoli et al., 2004).

Microtremor HVSR method has gained a great popularity also because it provides the sediments resonant frequency without knowing the thickness of sediments and their vertical S-waves velocity profile, the data that are otherwise needed to





numerically calculate the frequency. However, these data can be acquired only with relatively expensive geophysical investigations like seismic refraction or MASW methods and drilling. On the other hand microtremor method provides the results only when there is a relatively strong impedance contrast between sediments and bedrock (Ansal, 2004). If the knowledge on the sediments and bedrock physical properties is minor, the method should be first tested to prove that it is

effective in given geological conditions. This is especially important in case of very heterogeneous geological setting, like in the Idrija town. In this paper a study on the applicability of microtremor HVSR method in the Idrija town area is presented to evaluate if this method can contribute to the future seismic microzonation. High level seismic microzonation of the Idrija region is needed to support urbane planning, earthquake resistant design, protection of industrial and cultural heritage buildings and structures, and civil protection emergency planning. However, no data on S-waves velocity structure of

sediments and bedrock were collected so far in the Idrija town area using alternative methods (microtremor array methods, MASW, seismic refraction or velocity measurements in boreholes) to support seismic microzonation. This fact increased the motivation for the study based on microtremor HVSR method, because no a prior information on the velocity structure is needed for the interpretation of results. Moreover, recommendable future alternative investigations of the velocity structure using more expensive methods, could be more efficiently planned based on the microtremor HVSR results.

Geological setting in the Idrija town area (Fig. 1b) is rather complex and the region is cut by several faults which extends mainly in the Dinaric (NW-SE) direction, parallel to the major dextral strike-slip Idrija fault (Mlakar and Čar, 2009). The largest part of the area is built of Triassic dolomite, limestone and claystone. In-between are Carboniferous claystones and Permian sandstones. Upper Cretaceous limestone is present only in the SW part of investigated area. Quaternary alluvial

sediments extent along the both sides of the Idrijca river and in the lower part of the Nikova river. However, the scale of the map (1:25.000) prepared by Mlakar and Čar (2009) does not allow to show in details the whole extent of the soft sediments in the town area. More or less thick layer of weathered or highly fractured rock and thin alluvium can be expected also in other places. Another supposedly soft "soil" is related to the mining and smelting deposits put in place during centuries of mining activities (Fig. 2). Their extent roughly corresponds to the extent of Quaternary alluvial sediments, building a

situation which can make a seismic response quite complex. Therefore, in the Idrija town area, no simple relation is expected between the extent, homogeneity and thickness of supposedly soft sediments and seismic site amplification. The application of advanced quantitative investigation methods is thus needed to support any microzonation study in the area. Since microtremor HVSR studies were so far mainly performed in sedimentary basins of rather regular shape, where the relation between thickness of sediments and resonance frequency in more straight-forward, any study performed in more complex

geological setting is of wider scientific interest and contribute to the verification of the methodology.



## 2 Seismic hazard in the Idrija area

According to the official seismic hazard map of Slovenia for 475 years return period (Lapajne et al., 2001) Idrija area is in the 0.200 g peak ground acceleration zone, but not far (8 km) from the border with the 0.225 g zone. Since the highest seismic hazard is 0.250 g in the central and utmost NW Slovenia, Idrija area can be considered as relatively high seismic

hazard. The main question on the seismic hazard assessment in this area is the exact location of the strongest known historical earthquake in Slovenia which occurred in 1511 and had estimated magnitude of 6.8. It is supposed that it occurred in the Alps-Dinarides junction area somewhere in the western Slovenia, but the exact location and mechanism of this event are still debated (Fitzko et al., 2005). However, this earthquake is often referred in literature as Idrija earthquake. The main reason for this is that 500 years ago there were only few stone buildings (castles, churches etc.) in the whole W Slovenia and

in historical documents only sparse records on their earthquake damage are available (Košir and Cecić, 2011). This fact makes difficult to evaluate the macroseismic field, which is crucial to constrain the epicentre location and magnitude of the event. On the other hand, the Idrija mercury mine started its operation in 1495, a decade before the earthquake stroke. Although there were only two built (other were wooden) houses at that time in Idrija (church and mining inspector house), there are some secondary evidences on the damage (Košir and Cecić, 2011). The most important was the huge landslide

downstream from Idrija, which completely buried the Idrijca river. As a consequence the lake was formed and the whole mine was submerged as well as part of the town. It took seven years to retrofit the mine and enable its further operation. Another reason for naming the 1511 event as Idrija earthquake is the fact that the Idrija fault is the most important dextral strike-slip Dinaric fault in W Slovenia (Moulin et al., 2014). The town of Idrija is located approximately in the middle of its total length of 120 km. Through detailed geomorphological, structural-geological and paleoseismological studies it was

revealed that it has a potential to generate strong earthquakes (Moulin et al., 2016), although its recent seismicity (last century) is rather low (Živčić et al., 2011).

In the old official seismic hazard map of Slovenia showing MSK (Medvedev-Sponheuer- Karnik) intensities for 500 years return period (Ribarič, 1987), the supposed location of 1511 earthquake had great influence on distribution of zones with the

highest seismic hazard. According to this map Idrija is located in intensity VIII MSK zone, but 10 km north of it there is already intensity IX MSK zone. It is very obvious that such distribution of high intensity zones is controlled only by the 1511 event, listed in the applied seismic catalogue at certain coordinates which are very poorly constrained, because no other strong historical earthquakes are known in this area. Although intensity seismic hazard map are today not used any more in engineering design, because they are replaced by peak ground acceleration maps, a new intensity map for 475 years return

period was prepared in 2011 to be used in civil protection for emergency planning (Šket Motnikar and Zupančič, 2011). According to this map, which is much more smoothed in comparison to Ribarič (1987) map, Idrija is located in VIII EMS-98 (European Macroseismic Scale) zone and there is no IX EMS-98 intensity zone in vicinity.





In conclusion, according to all existing seismic hazard maps of Slovenia for 475 or 500 years return period, Idrija is located in relatively high seismic hazard zones: 0.200 g peak ground acceleration, VIII MSK or VIII EMS-98. Therefore, there is a great need to prepare also a comprehensive seismic microzonation map for the Idrija town area to enable thorough urbane planning, earthquake resistant design of buildings, protection of heritage buildings or structures and emergency planning.

5 **3 Methodology**

The microtremor method has been widely used for site effect studies in the last decade (Bard, 1999) although the theoretical basis of the Horizontal-to-Vertical Spectral Ratio (HVSR) analyses of the free-field microtremor measurements are still debated. Different theories on the content of body and surface waves in microtremors have been considered. More widely accepted is the "surface waves" explanation (Bard, 1999; Bonnefoy-Claudet et al., 2006), by which HVSR is related to the 10 ellipticity of Rayleigh waves, which is frequency dependent. HVSR therefore exhibits a sharp peak at the fundamental frequency of the sediments, when there is a high impedance contrast between the sediments and underlying bedrock. Criticism of the HVSR method was often related to the fact that there is no common practice for data acquisition and processing (Mucciarelli and Gallipoli, 2001) but some standards have been provided later (SESAME, 2004). Today it is widely accepted that the frequency of the HVSR peak reflects the main resonance frequency of the sediments. The main 15 advantages of the HVSR method are a straightforward estimate of the resonance frequency of sediments without knowing the geological and S-waves velocity structure of the subsurface, and relatively simple, low-cost measurements. This frequency can be used directly to assess the danger of soil-structure resonance or independently, because there is a trend to consider also the resonance frequency or period of sedimentary cover in different soil classifications for seismic microzonation (e.g. Ansal, 2004; Luzi et al., 2011). The use of microtremors was later extended to the study of dynamic 20 parameters of buildings, for instance for identification of their fundamental frequencies (Mucciarelli et al., 2001; Gallipoli et al., 2004; Boutin and Hans, 2008; Gosar et al., 2010).

**4 Microtremor measurements**

Microtremor measurements were performed in approximately 0.8 km$^2$ large area which extends across the whole urban area of Idrija on the both sides of the Idrijca and Nikova rivers (Figs. 1b and 2). Altogether 70 measurements were conducted 25 with the spacing of 50-200 m between measuring points. The locations were carefully selected to avoid as much as possible the influence of buildings, industrial facilities and traffic, although, in the built urban environment this was not always possible and the grid of measuring points is therefore quite irregular. Especially in the town centre the free-field space between houses was very limited, while in the industrial northern part selection of measurement points was limited due to restricted access to some industrial facilities.

30





Measurements were performed by two Tromino seismographs (Micromed, 2005) composed of three orthogonal electrodynamic velocity sensors, a GPS receiver, digitizer and recording unit with a flash memory card. All parts are integrated in a common case to avoid electronic and mechanical noise, which can be introduced by wiring between equipment parts. Good ground coupling on soft soil was obtained by using long spikes mounted at the base of the
seismograph. The sampling frequency was 128 Hz and the recording length at each point 20 minutes. The experimental conditions of microtremor measurements (e. g. Chatelain et al., 2008) were mainly favourable. The main difficulties arose from the low-frequency traffic and industrial noise.

The horizontal-to-vertical-spectral-ratio (HVSR) analysis was performed in the following way. Recorded time series were
visually inspected to identify possible erroneous measurements and stronger transient noise. Each record was then split into 30 s long non-overlapping windows, for which amplitude spectra in the range 0.1–64 Hz were computed using a triangular window with 5% smoothing and corrected for sensor transfer function. The HVSR was computed as the geometric average of both horizontal component spectra divided by the vertical spectrum for each window. Finally, the average HVSR function of all windows with the corresponding 95% confidence interval was computed. An example of the HVSR analysis for a
measurement at point Id46 is shown in Fig. 3. Amplitude spectral curves (Fig. 3a) shows a clear difference between both horizontal and the vertical component in a narrow frequency range. This difference results in a clear peak on the HVSR curve at 12.3 Hz with amplitude of 4.2 (Fig. 3b). Stability of the H/V spectral ratio within 20 minutes long record was analysed in 30 s long windows, as shown in Fig. 4 for a measurement at point Id3. From the colour-coded plot of HVSR functions for all windows (Fig. 4a), the windows including strong transient low-frequency noise were identified and
excluded from further computation (Fig. 4c). After removal of the noisy parts of the record the signal to noise ratio has improved in comparison to original HVSR curve (Fig. 4b), narrowing the 95% confidence interval, especially in the low-frequency part of the spectral ratio (Fig. 4d). A very clear and pronounced peak at 9.4 Hz, which has HVSR amplitude of 9.3, was obtained. For most of the measurements it was sufficient to exclude 2-8 windows (each 30 s long) from further computation, i.e. 5-20 % of the whole record length. However, on some locations low-frequency noise was very high and
persistent. In this case measurements were repeated at other time and the results compared. Temporal stability of the H/V spectral ratio was, afar exclusion of the most noisy windows, in general good. This is reflected in a relatively narrow 95% confidence interval for average HVSR functions which was obtained for most of the records (examples shown in Fig. 5). On the other hand, there were some locations (an example is Id5 on Fig. 6) with persistent low-frequency noise, also if the measurement was repeated later. Temporal stability of the H/V spectral ratio was in this case low and could preclude
determination of a resonance frequency. However, this problem was more pronounced at low-frequencies below 1 Hz, which are in general below the frequency range of engineering interest for low-rise buildings which prevail in the Idrija town.





## 5 Results of microtremor HVSR analyses

The HVSR analyses of free-field measurements showed that most of them (Fig. 5) fulfil the criteria for reliable measurements and a clear peak (SESAME, 2004). If sufficient number of criteria are fulfilled, the frequency of the peak is considered to be the fundamental frequency of sediments down to the first strong impedance contrast. The main reasons for

the failure of the above criteria are: a) high level of low-frequency noise during whole 20 minutes of recording, b) several peaks in a spectrum, or c) a flat spectral ratio. In cases in which the small amplitude of the HVSR peak caused failure to the criteria for a clear peak, we compared the results with adjacent measurements. If the frequencies of questionable peaks were comparable with the frequencies obtained at adjacent points, we kept them in the database. At the end, for 48 measuring point out of 70 it was possible to reliably define the peak frequency.

Several examples of the HVSR graphs are shown in Figs. 5 and 6. Locations of these measurements are shown in Figs. 1b and 2. In approximately one third of all measurements, very clear peaks were obtained showing the broad range of fundamental frequencies between 2.5 and 19.5 Hz. The temporal stability of the signal was moderate. In general 95% confidence interval of averaged curves is narrower at higher frequencies and wider at lower frequencies. This is mainly due

to the low-frequency industrial and traffic noise, which could not be completely removed by exclusion of windows with stronger transients. In some cases there is a sharp peak in the HVSR curve (Id2, Id37, Id54 and Id62 in Fig.5) which is rather symmetrical and has relatively large amplitude (5-7) and narrow 95% confidence interval. Such results are most desirable for derivation of sediments resonant frequency. However, they reflect the highest impedance contrast between sediments and the bedrock and thus the highest amplification of the seismic ground motion. Asymmetric shapes of HVSR curves with

additional side peaks, which have slightly lower amplitudes, are also common (Id13, Id30 and Id40 in Fig. 5). In general also in these cases it is possible to reliably determine the sediments resonance frequency although some criteria for a clear peak (SESAME, 2004) are not fulfilled. In some other cases the amplitude of the second peak has similar (Id9 in Fig. 5) or even higher amplitude (Id31 in Fig. 5) than the supposed main peak. However, these additional peaks occur at very high frequency (26 Hz in Id9 and 48 Hz in Id31) and cannot reflect the layers of seismological interest, because they are very thin.

It is even possible that some of these peaks at very high frequencies are artefacts of very noisy (monochromatic) conditions. Therefore, the frequency at which the main peak occurs is still a reliable estimate of the sediments resonance frequency. In some cases the level of low-frequency noise was rather high, which resulted in a wide 95% confidence interval at frequencies below 5 Hz (Id9 and Id40 in Fig. 5). In few cases (Id29 in Fig. 5) the 95% confidence interval is relatively wide in the whole spectral range and reflects very noisy measurement conditions in a wide range of frequencies. The stability of

the H/V spectral ratio was therefore lower and if the noise was persistent during the whole 20 min record, it was not possible to improve the signal-to-noise ratio by excluding the noisy windows as shown in Fig. 4.





In Fig. 6 several examples of more complex shaped HVSR curves or examples with a flat response are shown. Among complex shaped HVSR curves, more problematic for interpretation are examples with several peaks of similar amplitudes in the spectral range of seismological and earthquake engineering interest. At point Id6 (Fig. 6) there are four peaks with amplitude 3-4 in the frequency range 6-14 Hz, which are not well separated. This can reflect a multilayer setting above the

bedrock which includes alluvial sediments and mining deposits at the left bank of the Idrijca river (Fig. 2). However, impedance contrasts between layers seems to be relatively low. At point Id11 (Fig. 6) there is a series of peaks in the frequency range 9-20 Hz as well as a separated peak at 2.5 Hz. This can indicate a thin layer of sediments, complex topography (shown in Fig. 2) and probably another deeper geological interface. Interesting is a measurement at point Id20 (Fig. 6) with a clear, broad and high amplitude peak at around 4 Hz, on top of which are two small and narrow peaks of

similar amplitude of around 6.5. Although high amplitudes indicate large impedance contrast at the location of relatively thick mining deposits (Fig. 2), it is not possible to isolate only one frequency for eventual soil-structure resonance analysis and broader frequency range 3-6 Hz should be considered. For the measurement Id24 (Fig. 6) three low amplitude (3.0-3.5) peaks are characteristic in the frequency range 6-12 Hz showing again a multilayer setting composed of alluvial and mining deposits at the right bank of the Idrijca river (Figs. 1b and 2).

There were around ten measurements for which almost flat H/V spectral ratio was obtained. In theory this means no soft sediments or only small impedance contrast between the sediments and the bedrock. Consequently, no or very small amplification of seismic ground motion can be expected, which is a favourable condition in seismic microzonation. In given geological setting it was interesting to analyse, if such conditions are characteristic only for locations far from the valley

floors filled with alluvial and mining deposits, where the bedrock is supposed to be almost at the surface. However, for three examples shown in the Fig. 6, it is clear that they all reflect different geological conditions. Measurement Id15 is located at the middle of mining deposits (Fig. 2), but perhaps out of the extent of alluvial deposits. This can be an indication that old mining and smelting deposits can be very well compacted, reducing the impedance contrast with the bedrock. On the other hand, point Id25 is located outside of mining and alluvial deposits (Fig. 2) on the bedrock and thus logically does not shows

any impedance contrast. Measuring point Id45 is located in the western part of the town (Fig. 2) at the location where only thin weathered layer or colluvium is expected. In few cases measurements were dominated by prominent low frequency noise, which is reflected in high amplitudes in the H/V spectral ratio at frequencies below 1 Hz (Id5 in Fig. 6). This is usually not an indication of a very deep impedance contrast, but is an artefact of the poor stability of the three-component microtremor record at low frequencies, which could not be avoided by the exclusion of high-noise windows from the HVSR

computation, because it persisted through the whole measurement. Since the response in the frequency range of engineering interest (1-15 Hz) is low, also such measurement (Id5 in Fig. 6) can be considered as flat response.



## 6 Iso-frequency and HVSR peak amplitude maps

Distribution of established frequencies for 48 measuring point with sufficient HVSR response using 2 Hz wide intervals is shown in Fig. 7. From this graph it is clear that there is no systematic distribution of frequencies, which is usually observed in sedimentary basins with rather symmetrical shape. In such conditions the distribution of frequencies reflects the  gradual

thickening of the sediments towards the centre of the basin or buried valley. On the contrary, in the Idrija town area there is obviously no simple relation between topography of the Idrijca and Nikova river floors and established soft sediments resonance frequencies. This means that there are no rather uniform bodies of alluvial sediments along both rivers and that sediments thicknesses and physical properties can vary considerably within short distances. Such conditions are a challenge for any seismic microzonation, because it is difficult to interpolate or extrapolate data obtained at sparse discrete points in

the study area obtained for instance by drilling.

The data from 48 measuring points were used to prepare the iso-frequency map showing resonance frequencies of sediments (Fig. 8). The map was drawn using natural neighbour interpolation algorithm. The fundamental frequency of sediments shows a distribution in a range of 2.5-19.5 Hz. Very high frequencies (above 16 Hz) are limited to small areas located at the

slopes of the valleys, where only very thin layer of soft sediments or weathered material above the stiff bedrock is present. The largest high-frequency area is in the southern part of the town and extends in W-E direction. In the northern part there are two small high-frequency areas on the both sides of the Idrijca river, high above the river floor, where the bedrock is very shallow. Surprisingly, the lowest observed frequencies (2.5-4.0 Hz), which were established at many (11) points (Fig. 7) do not extent along booth rivers, where the largest thickness of alluvial sediments is expected, but in the central part of

investigated area, extending in W-E direction across both rivers (Fig. 8). There is also no correlation of the low-frequency area and the extent of mining and smelting deposits. For majority of the surveyed area relatively high-frequencies in the range 8.0-14.0 Hz are characteristic, also along booth rivers and in areas with artificial deposits. This is an indication that the sediments or deposits are relatively thin and/or rather well compacted.

The HVSR peak amplitudes map is shown in Fig. 9. The amplitudes of microtremor HVSR peaks are mainly in the range 3-6, only in few cases they reach higher values. The amplitude of the peak is related to the impedance contrast between sediments and the bedrock, but in principle it cannot be used for quantitative estimation of this contrast or site amplification (SESAME, 2004). There is no visible correlation between certain frequency range (Fig. 8) and distribution of higher or lower peak amplitudes (Fig. 9), as well as with the extent of alluvial sediments or artificial deposits.


Comparison of both maps derived from microtremor HVSR measurements (Figs. 8 and 9) with the geological map (Fig. 1b) and distribution of mining and smelting deposits (Fig. 2) shows that in the Idrija town area there is no clear relation to the extent of sediments that are usually characterized as "soft" in seismic microzonation. In such conditions, a very extensive





program of geophysical investigations (and/or drilling) using for example seismic refraction and MASW methods will be needed to derive S-waves velocity profiles and the depth down to the "seismological" bedrock and numerically calculate the site amplification and resonance frequency in the whole town area. To some extent, the microtremor HVSR method has advantage, because it provides sediments main resonance frequency without knowing the S-waves velocity profile and their

thickness. Iso-frequency map can be used also to directly assess the danger of soil-structure resonance. Recently, large number of masonry buildings was surveyed with microtremor method in five Slovenian towns (Gosar, 2012) to derive main building frequencies. Statistical analysis of these results versus number of floors was performed to generalize identification of potential soil-structure resonance. Taking into account that two- and three-floors masonry buildings prevail in Slovenian towns, their frequency range was established as 5.6-11.1 Hz. The possible occurrence of the soil-structure resonance should

be therefore sought especially in this frequency range also in the Idrija town where similar buildings prevail. From the iso-frequency map shown in Fig. 8 it is clear that 5.6-11.1 Hz frequency range occupies a large part of the surveyed town area and therefore, soil-structure resonance could be a serious issue also in Idrija. This should be further investigated by microtremor measurements within individual buildings and direct comparison of the results with the free-field data. For generalization of results to the characteristical set of buildings, it would be necessary to establish relations between buildings

height and their fundamental frequencies (longitudinal, transversal). Whereas several such investigations were performed in Europe recently for reinforced-concrete buildings (e.g. Gallipoli et al., 2010), there is still lack of such studies for masonry buildings, which prevail in the Idrija town. However, first the free-field resonance frequency map was prepared to support further soil-structure resonance analyses.

Another application of the derived iso-frequency map is to support soil classification, because recent investigations (Luzi et al., 2011) have shown that fundamental frequency of sediments can be used together with the average S-waves velocity to improve classifications according to different building codes. However, not many examples of such application of microtremor HVSR results were published so far.

### 7 Conclusions

Microtremor HVSR investigations performed in the Idrija town area have shown that this method is applicable to support seismic microzonation in given geological conditions. The main results of the study are iso-frequency and HVSR peak amplitude maps of the soft sediments. Although part of the town is built on alluvial sediments of Nikova and Idrijca rivers and on artificial mining and smelting deposits that were put in place through centuries of mining activity, there is no clear correlation between extent and supposed thickness of sediments and distribution of resonance frequencies or HVSR peak

amplitudes. This can be explained by rather complex and uneven geometry of sediments or deposits and different level of their compaction (stiffness). At approximately one third of all measured locations it was not possible to derive resonance frequency, because of a flat or nearly flat HVSR curves. This is of course not a drawback of the microtremor method,



because a flat HVSR curve indicates a small or no impedance contrast between sediments and the "seismological" bedrock and thus no amplification of ground motion. On locations where the frequency and the peak HVSR amplitude were derived, the microtremor method could have advantages in comparison to numerical solutions based on geophysical investigations and/or drilling. In a complex geological setting a very large number of geophysical measurements is needed to obtain

representative data. This can be very expensive and difficult to realize in densely built environment, where the free space to conduct for example active seismic refraction or MASW geophysical measurements is very limited and also very sensitive to usually high level of noise. On the other hand, microtremor method can be effectively applied also in such conditions, which are unfavourable for geophysical investigations. The study performed has shown that the microtremor HVSR method could be successfully applied also in complex geological setting and thus represents an important validation of the methodology,

which is of wider scientific interest, especially because most of the published studies are from sedimentary basins of rather regular shape. Moreover, recommendable future alternative investigations of the velocity structure using more expensive methods could be more efficiently planned based on the microtremor HVSR results.

Sediments main resonance frequency ($f_0$) derived from microtremor HVSR measurements has two basic applications in the

seismic microzonation. It can be used directly, together with the data on the building fundamental frequencies, to assess the possible occurrence of soil-structure resonance (e.g. Gallipoli et al., 2004; Gosar, 2012). Secondly, it can be a complement to the average S-wave velocity in the upper 30 m ($Vs_{,30}$) as proposed by Ansal (2004) and Luzi et al. (2011). Most of the seismic codes make use of the $Vs_{,30}$ to discriminate soil categories, although some doubts exist about the capability of $Vs_{,30}$ to predict actual amplification of sediments. Luzi et al. (2011) showed that there is a significant reduction of the standard

deviation associated to the ground motion prediction when the classification is based on the couple of variables $Vs_{,30}$-$f_0$. However, further investigations on these issues are needed and any successful application of the microtremor HVSR method in complex conditions like in the Idrija town is a valuable scientific contribution to achieve the goals. In the Idrija town area both applications are feasible, because there is a great need to prepare a high level quantitative microzonation in the future and also to directly analyse the danger of soil-structure resonance for individual buildings, including cultural heritage mining

and other structures protected by UNESCO, to prevent them from earthquake hazard.

**Acknowledgements**

The study was realized with the support of the research program P1-0011 financed by Slovenian Research Agency.

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




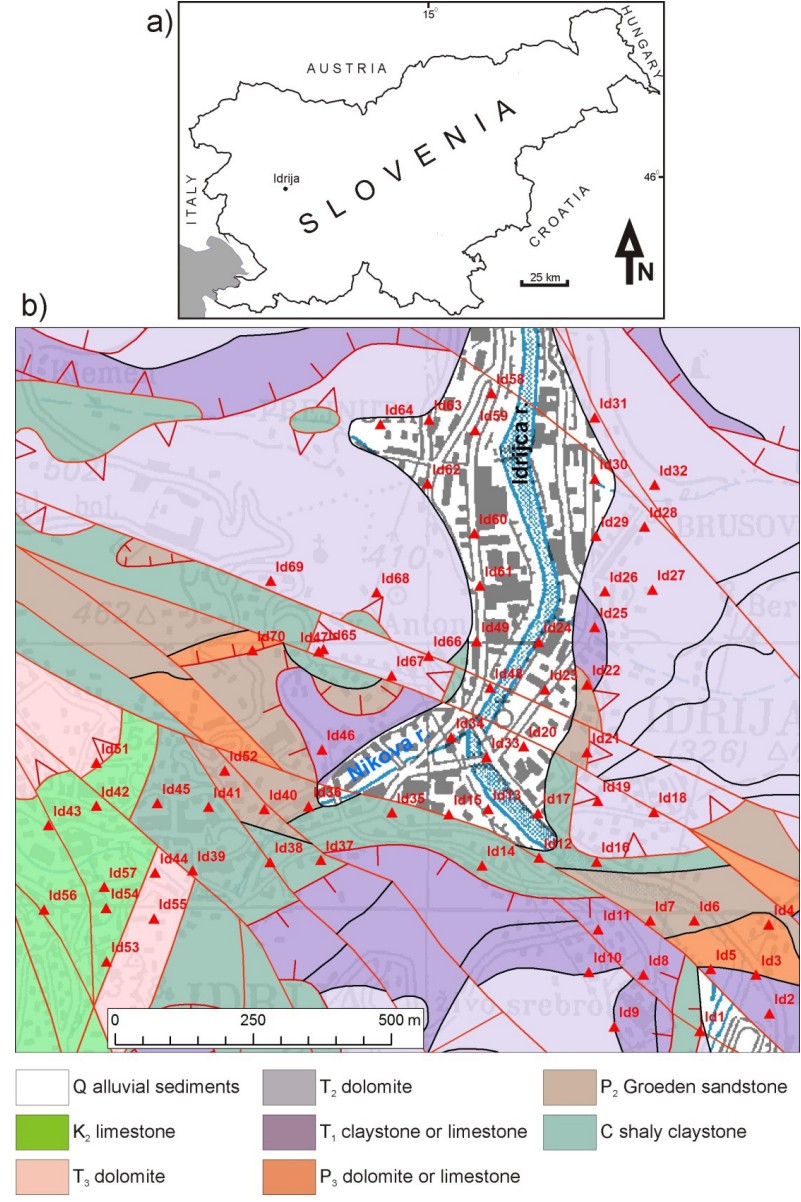

**Figure 1: a) Location of the Idrija town in Slovenia. b) Geological map of the Idrija town area (after Mlakar and Čar, 2009). Red triangles indicate points of microtremor measurements.**





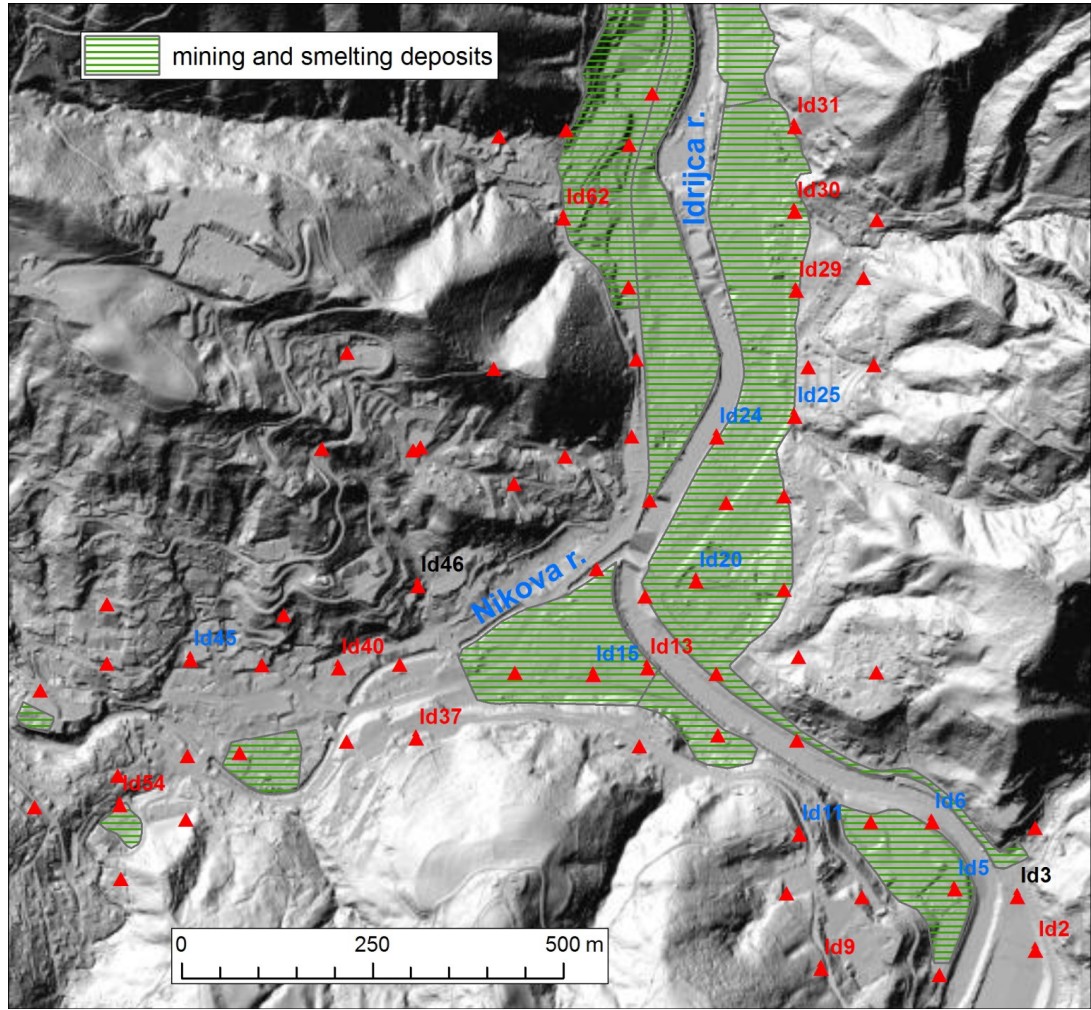

**Figure 2: Position map of microtremor measurements (red triangles) in the Idrija town area. Red labelled points indicate examples of HVSR analyses shown in Fig. 5, blue labelled points examples shown in Fig. 6 and black labelled points examples shown in Figs. 3 and 4. Extent of mining and smelting deposits is shown (data provided by Idrija Municipality). Basemap shaded relief was derived from LiDAR 1 m resolution Digital Elevation Model (Ministry of the Environment and Spatial Planning, portal e-Vode).**



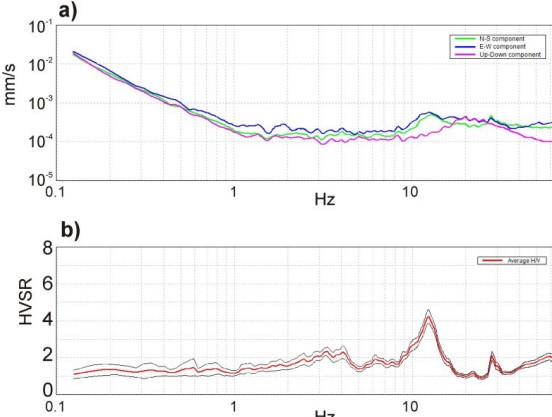

**Figure 3: An example of microtremor Horizontal-to-Vertical Spectral Ratio (HVSR) analysis at point Id46. (a) Three amplitude spectral curves clearly shows the difference between both horizontal (N-S, E-W) and the vertical (up-down) component in a narrow frequency range. (b) Observed difference results in a clear peak on the HVSR curve at 12.3 Hz. Thin lines represent the**
5  **95% confidence interval.**





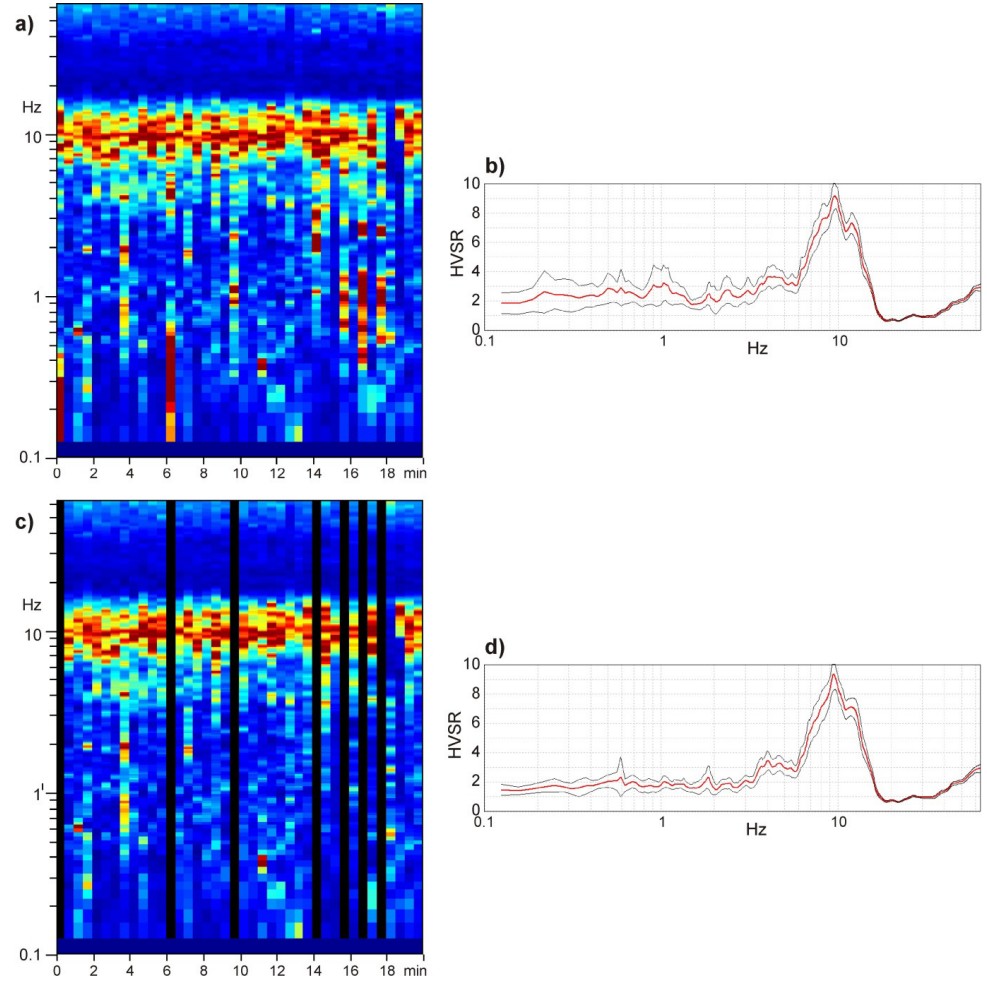

**Figure 4: a) Stability of the H/V spectral ratio within 20 minutes long record at point Id3 analysed in 30 s long windows. b) Corresponding average HVSR curve for all windows. c) Removal of several noisy windows from the record resulted in d) better signal to noise ratio in HVSR curve.**





**Figure 5: Examples of microtremor measurements on points where a clear HVSR response was obtained. Locations of measurements are shown in Fig. 2. Thin lines represent the 95% confidence interval.**



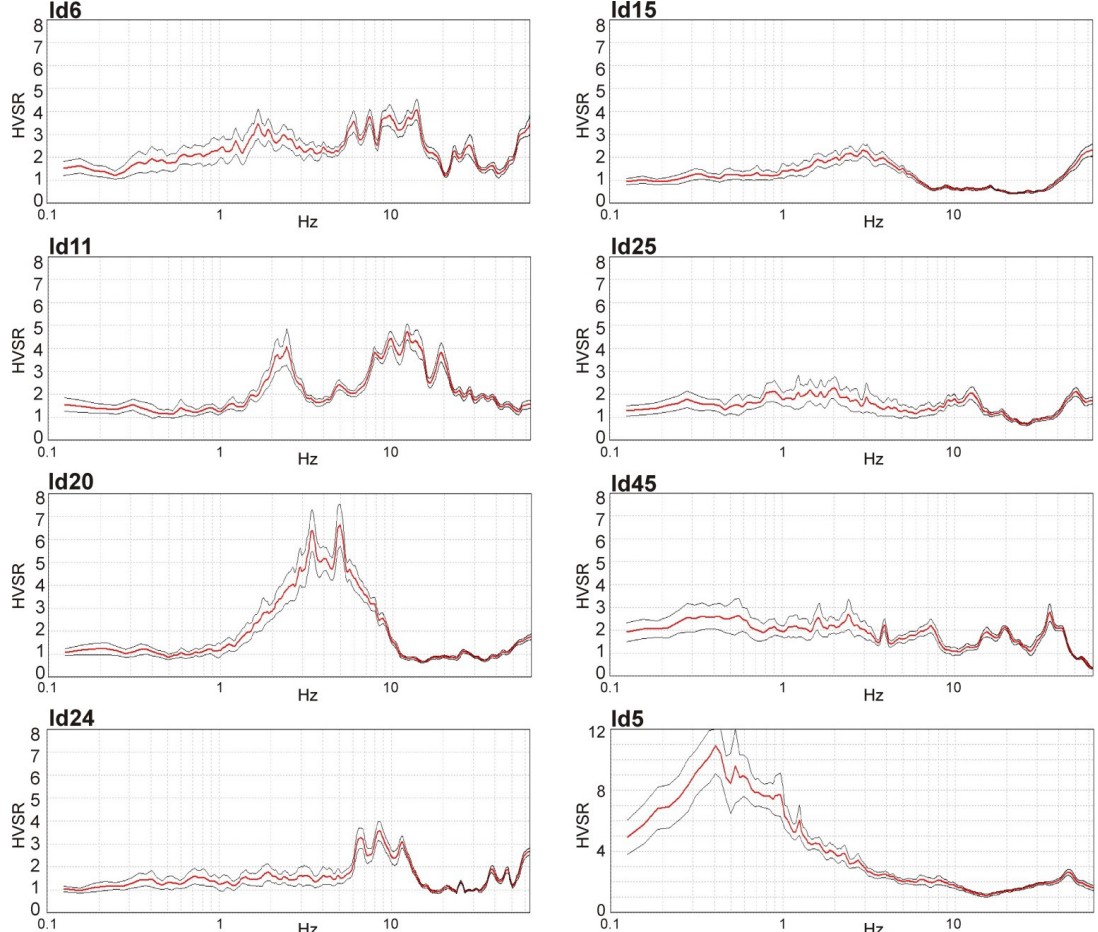

**Figure 6: Left side: Examples of microtremor measurements with several peaks in HVSR curve. Right side: Examples of microtremor measurements with no HVSR response or pronounced low-frequency noise (Id5). Locations of measurements are shown in Fig. 2. Thin lines represent the 95% confidence interval.**





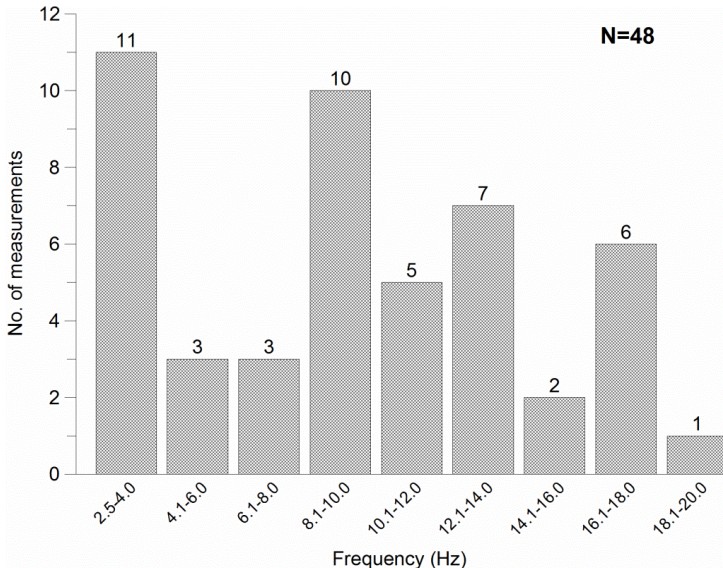

**Figure 7: Distribution of sediments main frequency derived from microtremor HVSR analysis at 48 points with sufficient response.**



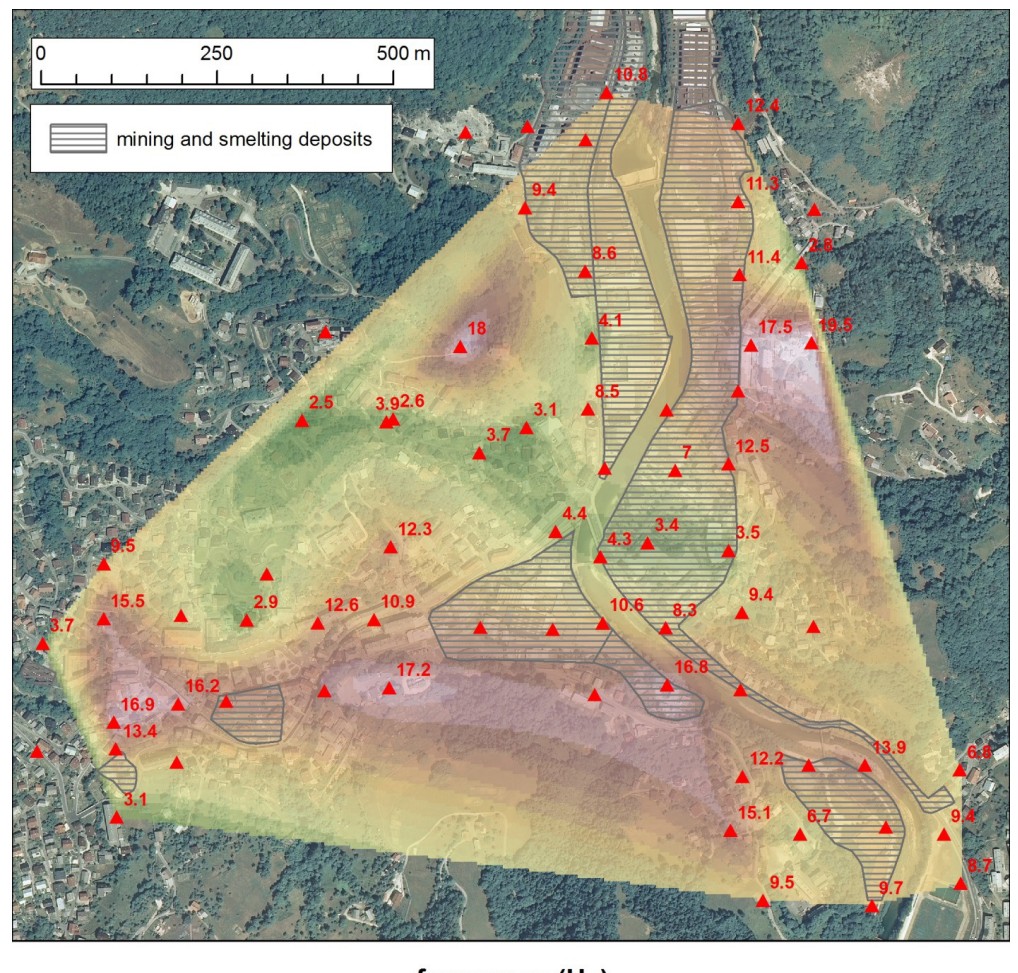

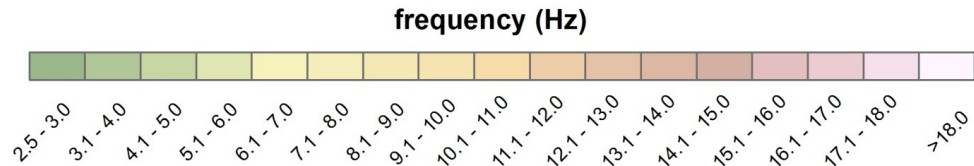

**Figure 8: Map of the sediments main resonance frequency in the Idrija town area derived from microtremor measurements. Triangles indicate points of all measurements. Corresponding labels indicate the resonance frequency for locations with sufficient HVSR response. Basemap is Digital Ortho Photo image (Surveying and Mapping Authority of the Republic of Slovenia).**





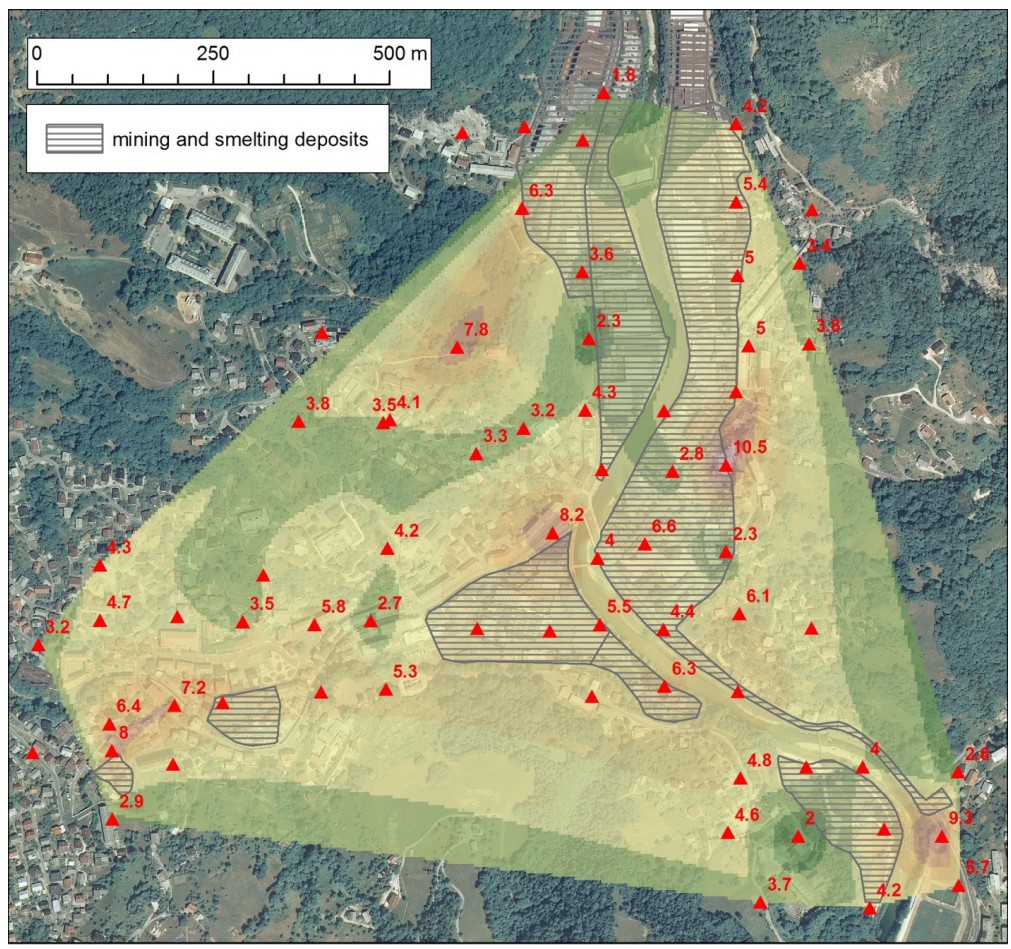

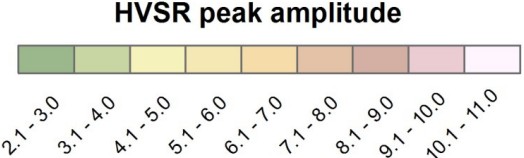

**Figure 9: Map of microtremor HVSR peak amplitudes in the Idrija town area. Triangles indicate points of all measurements. Corresponding labels indicate the peak amplitudes for locations with sufficient HVSR response. Basemap is Digital Ortho Photo image.**

