# Peer review of "Study on the applicability of the microtremor HVSR method to support seismic microzonation in the town of Idrija (W Slovenia)"

_Natural Hazards and Earth System Sciences, 2016_

## Referee Comment (RC1) · Anonymous Referee #1 · 20 Feb 2017

General comments The effort of this paper consists in the huge amount of ambient noise recordings performed in the Idrija town area. Notwithstanding this large contribution of measurements, it was not possible to characterize the lito-stratigraphic site characteristics. As the author himself has pointed out the geological setting is complex and there is no correlation between site effects and geological map, even where it was to be expected (alluvial sediments of Nikova and Idrijca rivers and on artificial mining and smelting deposits).

Specific comments The geological section requires more geological and stratigraphic details, I suggest to add the geological cross section and details about litological units. I believe that the resonance peaks at $f_0 > 12$ Hz should not be considered for the es-

timation of the iso-frequency map because they are attributable at so thin soil layers (1-2m) not influent for site condition. Therefore, the resonance peaks useful for estimation of the iso-frequency map are only 32. Moreover, the presence of peaks at so close and narrow frequencies (e.g. Id24, Id6, Id13 etc..) are not attributable to a multilayer setting above the bedrock because probably some are spurious spikes. The shapes of these spikes are more visible in the Fourier spectra, I suggest to analyze the ambient noise measures using a triangular window with 10% smoothing. I suggest to group the measures according the same lito-stratigraphic condition so as to highlight if the geological condition induces a similar site amplification behavior. It is unclear how two measurements performed on the same geological unit have HVSR functions completely different, for example the Id20 measure close to Id25 one are on the same alluvial sediment but the HVSR shapes are completely different.

Technical corrections Include the orientation of the map shown in Fig. 1b, Fig. 2, Fig. 8 and Fig. 9. In the iso-frequency map I suggest to contour better the areas, the south and east areas have no data then the frequency interpolations are wrong.

---

## Referee Comment (RC2) · Anonymous Referee #2 · 3 Apr 2017

The paper presents a method which has already been extensively studied and presented by many authors in the past decades. There is no new scientific information in this paper. However, the results are clearly presented and well documented.

The English should be revised by a English speaking person, as there are many errors remaining throughout the text.

In the References section, each reference should start by a tabulation, as it is, otherwise, not possible to separate them when looking for a particular reference.

A main point should be kept in mind throughout the paper : in order to be useful for engineers who make the design or control of buildings against earthquake, a microzonation study should provide response spectra associated to each micro-zone. Alone, resonance frequency and amplitude maps are of no use for the earthquake engineering community, and thus of no use for seismic risk reduction. The resonance frequency is useful to calibrate the Vs profile of a site, very uncertain even when Vs measurements are conducted. Resonance frequency measurements only are the very first step of a complete microzonation study that should lead to the definition of site specific response spectra for engineers.

---

## Author Comment (AC1) · 7 Apr 2017

Response to Referee #1 comments

Comm.: The geological section requires more geological and stratigraphic details, I suggest to add the geological cross section and details about lithological units.

Response: More details on the lithological units of alluvial sediments, which are relevant for seismic site amplification, were added to the last paragraph of the Introduction. No detailed cross-section of these sediments from drilling or geophysical investigations is available to be included, this is explained in the text. This fact was one of the main motivation for presented microtremors study, because only in this way resonance frequencies could be obtained without detailed knowledge on sediments thickness and S-velocity structure. Additional explanation on this was added to the text.

Comm.: I believe that the resonance peaks at f0>12 Hz should not be considered for the estimation of the iso-frequency map because they are attributable at so thin soil layers (1-2m) not influent for site condition.

Response: I prepared a map taking into account only resonance peaks with f0<12 Hz. However, the result was not satisfactory, because we cannot consider HVSR curves with higher peak frequencies (f0>12 Hz) as a flat response although they are above the engineering interest. Some very clear peaks were obtained at higher frequencies (for example Id37 and Id54 in Fig. 5). Omitting these points from contouring resulted in values in the map (at this points) obtained by interpolation, which are far from measured and thus wrong. This can lead to erroneous assessment of potential soil-structure resonance at particular location.

Comm.: The presence of peaks at so close and narrow frequencies (e.g. Id24, Id6, Id13 etc..) are not attributable to a multilayer setting above the bedrock, because probably some are spurious spikes. The shapes of these spikes are more visible in the Fourier spectra, I suggest to analyse the ambient noise measures using a triangular window with 10% smoothing.

Response: A discussion on the nature of several peaks was added to the text. For Id6 and Id24 the influence of multilayer setting is now explained only as additional factor which can have impact on the shape of HVSR curve, because both measurements are located in an area where both artificial deposits and alluvial sediments are expected. For Id13 it is explained as spurious peak which has no influence on determination of a resonance frequency. All measurements were analysed as Fourier spectra (one example is shown in Fig. 3) and as HVSR with different level of smoothing, including 10% smoothing. Additional explanation on this was added to the text.

Comm.: I suggest to group the measures according the same lito-stratigraphic con-

dition so as to highlight if the geological condition induces a similar site amplification behaviour. It is unclear how two measurements performed on the same geological unit have HVSR functions completely different, for example the Id20 measure close to Id25 one are on the same alluvial sediment but the HVSR shapes are completely different.

Response: Text was improved to highlight how similar geological conditions induces a similar site amplification behaviour. However, it is clearly described that site amplification cannot be simply correlated to the surface geology, due to very heterogeneous geotechnical properties of artificial and alluvial deposits. Unknown shallow subsurface structure (S-velocity and density distribution) was the main motivation for a study based on microtremors which does not require a-priori knowledge on it to derive resonance frequency. Additional explanation was added to the text to highlight this facts. Measurement Id20 is located on artificial and deposits (shown in Figs. 1 and 2) which are reflected in a clear HVSR peak. On the other hand measurement Id25 is located on a bedrock (this is visible in Figs. 1 and 2) which is reflected in a flat HVSR response.

Comm.: Include the orientation of the map shown in Fig. 1b, Fig. 2, Fig. 8 and Fig. 9. In the iso-frequency map I suggest to contour better the areas, the south and east areas have no data then the frequency interpolations are wrong.

Response: The orientation mark (North arrow) was added to all maps as suggested. The iso-frequency map (Fig. 8) was contoured in a better way to consider missing data.

---

## Author Comment (AC2) · 7 Apr 2017

Response to Referee #2 comments

Comm.: The English should be revised by a English speaking person, as there are many errors remaining throughout the text.

Response: The final version of the manuscript will be corrected by professional English proof-reader.

Comm.: In the References section, each reference should start by a tabulation, as it is, otherwise, not possible to separate them when looking for a particular reference.

Response: The Reference section will be corrected as suggested and according to the instructions for authors.

Comm.: A main point should be kept in mind throughout the paper: in order to be useful for engineers who make the design or control of buildings against earthquake, a microzonation study should provide response spectra associated to each micro-zone. Alone, resonance frequency and amplitude maps are of no use for the earthquake engineering community, and thus of no use for seismic risk reduction. The resonance frequency is useful to calibrate the Vs profile of a site, very uncertain even when Vs measurements are conducted. Resonance frequency measurements only are the very first step of a complete microzonation study that should lead to the definition of site specific response spectra for engineers.

Response: The purpose of the study will be additional described and justified throughout the paper. I fully agree with the comment that resonance frequency measurements are only the first step of a microzonation, but at the same time a very important one for given conditions in the town of Idrija. I believe that this is clear enough from different parts of the manuscript as:

in the Title: . . .. to support seismic microzonation . . ..

in Abstract (last three sentences): The importance of microtremor method is therefore even greater, because it enables direct estimation of the resonance frequency without knowing the internal structure and physical properties of the shallow subsurface. The results of this study can be used directly in analyses of possible occurrence of soil-structure resonance of individual buildings, including important cultural heritage mining and other structures protected by UNESCO. Second application of the derived free-field iso-frequency map is to support soil classification according to the recent trends in building codes.

in Introduction (last part): In the Idrija town area, no simple relation is expected between the extent, homogeneity and thickness of supposedly soft sediments and seismic site

amplification. The application of advanced quantitative investigation methods is thus needed to support any microzonation study in the area. Since microtremor HVSR studies were so far mainly performed in sedimentary basins of rather regular shape, where the relation between thickness of sediments and resonance frequency is more straight-forward, any study performed in more complex geological setting is of wider scientific interest and contribute to the verification of the methodology.

in Conclusions (last paragraph): Sediments main resonance frequency (f0) derived from microtremor HVSR measurements has two basic applications in the seismic microzonation. It can be used directly, together with the data on the building fundamental frequencies, to assess the possible occurrence of soil-structure resonance (e.g. Gallipoli et al., 2004; Gosar, 2012). Secondly, it can be a complement to the average S-wave velocity in the upper 30 m (Vs,30) as proposed by Ansal (2004) and Luzi et al. (2011). Most of the seismic codes make use of the Vs,30 to discriminate soil categories, although some doubts exist about the capability of Vs,30 to predict actual amplification of sediments. Luzi et al. (2011) showed that there is a significant reduction of the standard deviation associated to the ground motion prediction when the classification is based on the couple of variables Vs,30-f0. However, further investigations on these issues are needed and any successful application of the microtremor HVSR method in complex conditions like in the Idrija town is a valuable scientific contribution to achieve the goals. In the Idrija town area both applications are feasible, because there is a great need to prepare a high level quantitative microzonation in the future and also to directly analyse the danger of soil-structure resonance for individual buildings, including cultural heritage mining and other structures protected by UNESCO, to prevent them from earthquake hazard.

---

## Author Response (AR1)

**Response to Referee #1 comments**

Manuscript NHESS-2016-405 **Study on the applicability of microtremor HVSR method to support seismic microzonation in the town of Idrija (W Slovenia)**

**Referee #1**

**Comm.:** The geological section requires more geological and stratigraphic details, I suggest to add the geological cross section and details about lithological units.

10 **Response:** A detailed description of the lithological units, which are relevant for seismic site amplification, was added at the end of the Introduction chapter. No detailed cross-section of these sediments from drilling or geophysical investigations is available to be included, and this is explained in the text. This fact was one of the main motivation for presented microtremors study, because only in this way resonance frequencies could be obtained without detailed knowledge on sediments thickness and S-velocity structure. Additional explanation on this was added to the text.

**Comm.:** I believe that the resonance peaks at f0>12 Hz should not be considered for the estimation of the iso-frequency map because they are attributable at so thin soil layers (1-2m) not influent for site condition.

**Response:** I prepared a map taking into account only resonance peaks with f0<12 Hz. However, the result was not satisfactory, because we cannot consider HVSR curves with higher peak frequencies (f0>12 Hz) as a flat response, although

20 they are above the engineering interest. Some very clear peaks were obtained at higher frequencies (for example Id37 and Id54 in Fig. 5). Omitting these points from contouring resulted in values in the map (at this points) obtained by interpolation, which are far from measured and thus wrong. This can lead to erroneous assessment of potential soil-structure resonance at particular location. However, the areas in the south and east with no data were excluded from contouring for a more realistic map. A detailed description on data selection and contouring the maps was added to the chapter Iso-frequency and HVSR

25 peak amplitude maps.

**Comm.:** The presence of peaks at so close and narrow frequencies (e.g. Id24, Id6, Id13 etc..) are not attributable to a multilayer setting above the bedrock, because probably some are spurious spikes. The shapes of these spikes are more visible in the Fourier spectra, I suggest to analyse the ambient noise measures using a triangular window with 10% smoothing.

30 **Response:** A discussion on the nature of several peaks was added to the text. In most cases, the side peaks were recognized as spurious peaks resulting from industrial noise. For Id6 and Id24 the possible influence of multilayer setting is now described only as an additional factor which can have impact on the shape of HVSR curve, because both measurements are located in an area where both artificial deposits and alluvial sediments are expected. For Id13 it is explained as spurious peak which has no influence on determination of a resonance frequency. All measurements were analysed as Fourier spectra (one

example is shown in Fig. 3) and as HVSR with different level of smoothing, including 10% smoothing. Additional explanation on this was added to the text.

**Comm.:** I suggest to group the measures according the same lito-stratigraphic condition so as to highlight if the geological condition induces a similar site amplification behaviour. It is unclear how two measurements performed on the same geological unit have HVSR functions completely different, for example the Id20 measure close to Id25 one are on the same alluvial sediment but the HVSR shapes are completely different.

**Response:** A long paragraph on comparison of the results based on lithology was added to the Results chapter with detailed description how similar geological conditions influence amplification behaviour. In addition, it is now more clearly emphasized that site amplification cannot be simply correlated to the surface geology, due to very heterogeneous geotechnical properties of artificial and alluvial deposits. Unknown shallow subsurface structure (S-velocity and density distribution) was the main motivation for a study based on microtremors which does not require prior knowledge on it to derive resonance frequency. Additional explanation was added to the text to highlight this facts. Measurement Id20 is located on artificial and deposits (shown in Figs. 1 and 2) which are reflected in a clear HVSR peak. On the other hand measurement Id25 is located on a bedrock (this is visible in Figs. 1 and 2) which is reflected in a flat HVSR response. This explains why the HVSR shapes of both measurements are different.

**Comm.:** Include the orientation of the map shown in Fig. 1b, Fig. 2, Fig. 8 and Fig. 9. In the iso-frequency map I suggest to contour better the areas, the south and east areas have no data then the frequency interpolations are wrong.

**Response:** The orientation mark (North arrow) was added to all maps as suggested. The iso-frequency map (Fig. 8) was contoured in a better way to consider missing data.

**Response to Referee #2 comments**

Manuscript NHESS-2016-405 **Study on the applicability of microtremor HVSR method to support seismic microzonation in the town of Idrija (W Slovenia)**

**Referee #2**

**Comm.:** The English should be revised by a English speaking person, as there are many errors remaining throughout the text.

10  **Response:** The final version of the manuscript was corrected by professional English proof-reader.

**Comm.:** In the References section, each reference should start by a tabulation, as it is, otherwise, not possible to separate them when looking for a particular reference.

**Response:** The Reference section was corrected as suggested and according to the instructions for authors.

**Comm.:** A main point should be kept in mind throughout the paper: in order to be useful for engineers who make the design or control of buildings against earthquake, a microzonation study should provide response spectra associated to each micro-zone. Alone, resonance frequency and amplitude maps are of no use for the earthquake engineering community, and thus of no use for seismic risk reduction. The resonance frequency is useful to calibrate the Vs profile of a site, very uncertain even

20  when Vs measurements are conducted. Resonance frequency measurements only are the very first step of a complete microzonation study that should lead to the definition of site specific response spectra for engineers.

**Response:** The purpose of the study was additional described and justified throughout the paper. I fully agree with the comment that resonance frequency measurements are only the first step of a microzonation, but at the same time a very important one for given conditions in the town of Idrija. I believe that this is clear enough from different parts of the

25  manuscript as:

in the Title: …. to support seismic microzonation ….

[revised manuscript text omitted]